# The Role of Metabolic Syndrome in Psoriasis Treatment Response: A One-Year Comparative Analysis of PASI Progression

**DOI:** 10.3390/diagnostics14242887

**Published:** 2024-12-23

**Authors:** Maria-Lorena Mustață, Mihaela Ionescu, Lucrețiu Radu, Carmen-Daniela Neagoe, Roxana-Viorela Ahrițculesei, Radu-Cristian Cîmpeanu, Daniela Matei, Anca-Maria Amzolini, Maria-Cristina Predoi, Simona-Laura Ianoși

**Affiliations:** 1Doctoral School, University of Medicine and Pharmacy of Craiova, 200349 Craiova, Romania; umlorena@yahoo.com (M.-L.M.); roxana.blendea@gmail.com (R.-V.A.); cimpeanu_r@yahoo.com (R.-C.C.); 2Department of Medical Informatics and Biostatistics, University of Medicine and Pharmacy of Craiova, 200349 Craiova, Romania; 3Department of Hygiene, University of Medicine and Pharmacy of Craiova, 200349 Craiova, Romania; lucretiu.radu@gmail.com; 4Department of Internal Medicine, Faculty of Medicine, University of Medicine and Pharmacy of Craiova, 200349 Craiova, Romania; dananeagoe2014@gmail.com; 5Department of Physical and Rehabilitation Medicine, University of Medicine and Pharmacy of Craiova, 200349 Craiova, Romania; mateidana30@yahoo.com; 6Department of Internal Medicine, Medical Semiology, University of Medicine and Pharmacy of Craiova, 200349 Craiova, Romania; anca.amzolini@umfcv.ro; 7Department of Morphology, Faculty of Medicine, University of Medicine and Pharmacy of Craiova, 200349 Craiova, Romania; predoi.cristina@yahoo.com; 8Department of Dermatology, Faculty of Medicine, University of Medicine and Pharmacy of Craiova, 200349 Craiova, Romania; simonaianosi@hotmail.com

**Keywords:** psoriasis, metabolic syndrome, obesity, systemic inflammation, biologic therapies, PASI score, leptin

## Abstract

Background/Objectives: Psoriasis is a chronic dermatological condition with systemic implications, especially with metabolic syndrome (MS). This study evaluated the vicious cycle where obesity and MS exacerbate systemic inflammation that complicates the efficacy of psoriasis therapies by examining the PASI score over a one-year period. Patients were classified into two subgroups: those with psoriasis alone (PSO) and those with both psoriasis and metabolic syndrome (PSO-MS). Methods: A total of 150 patients, half of whom also concomitantly presented with metabolic syndrome, received biologic therapies comprising anti-IL-17, anti-IL-23, and anti-TNF-a, or methotrexate, with PASI scores assessed at baseline and at 3, 6, and 12 months. Results: All treatments showed significant reductions in PASI; however, patients with PSO showed more marked reductions in PASI score than those in the PSO-MS group. Anti-IL-17 treatments produced the greatest sustained long-term improvements, whereas anti-IL-23 produced prompt early improvements. Increases in BMI and leptin concentrations were associated with a modest rate of reduction in PASI score, underlining the impact of obesity and metabolic dysfunction on treatment efficacy. Conclusions: This study highlights the importance of managing comorbidities such as MS in the treatment of psoriasis, as the interplay between systemic inflammation and metabolic health further complicates therapeutic outcomes.

## 1. Introduction

Psoriasis is a long-lasting and dynamic inflammatory dermatological condition known for its significant morbidity and disfiguring effects [1]. It affects approximately 2–3% of the worldwide population and typically appears on the elbows, knees, and lower back, presenting well-defined erythematous plaques covered in silvery scales [2].

The complex pathogenic pathway of psoriasis involves numerous immune molecules and immunocytes [3,4]. Activated T cells initiate a sequence of cellular and molecular reactions culminating in psoriatic lesions [5,6]. While the autoimmune nature of the disease is widely recognized and treatments targeting T cell activity have been effective, the precise mechanisms and origins of the abnormal T cells involved in psoriasis remain uncertain [7,8]. Systemic inflammation is mostly involved in the development of this chronic inflammatory disease, which is linked to several comorbidities such as metabolic syndrome and thyroid and liver diseases [9].

On a global scale, psoriasis affects approximately 125 million people. It is more prevalent among Caucasians, and while earlier studies suggested no significant difference in its impact between men and women, more recent studies indicate that men tend to have more severe disease [10,11], while women report a greater psychosocial burden and a higher frequency of joint involvement and comorbid conditions [12,13].

The estimated incidence of psoriasis in the populations of Northern Europe and Scandinavia ranges between 1.5% and 3%. However, it is notably less common among Chinese, West Africans, and North Americans. Among North American and South American natives, as well as Aboriginal Australians, psoriasis is sporadic [14]. These discrepancies are likely to be the result of genetic and environmental factors [15]. Several research studies suggest that psoriasis is a genetic condition with a polygenic distribution and variable impact, as shown in population-based and twin studies [16].

Psoriasis can present at any age and often persists throughout an individual’s lifetime [17]. In several reports, the median age of onset of psoriasis is 33 years, with 75% of patients developing the condition before the age of 46 [18]. It has been hypothesized that the onset of psoriasis follows a bimodal distribution, with a peak at 16–22 years of age and a second peak at 57–60 years of age or older. Women generally experience the condition at a slightly younger age than men [19].

Psoriasis is nowadays acknowledged as a systemic disorder with connections to various comorbidities, including metabolic syndrome (MS), obesity, and cardiovascular disease, despite its classification as a dermatological condition [20]. These connections are significantly influenced by the persistent systemic inflammation associated with psoriasis, shifting the focus from viewing psoriasis solely as a skin disease to recognizing it as a systemic inflammatory condition [21,22]. The common inflammatory pathways of these conditions perpetuate a vicious cycle of systemic inflammation. Adipose tissue, in particular visceral fat, is far from being just an energy store, it functions as a complex endocrine organ secreting numerous molecules called adipokines. These have important effects on metabolism, inflammation, and the immune system [23]. With the accumulation of adipose tissue, especially visceral fat, as in obesity, there is a dysregulation of adipokine secretion which results in a chronic inflammatory state. Leptin, IL-6, and TNF-α are the main proinflammatory cytokines secreted by adipose tissue. The potent inflammatory cytokines TNF-a and IL-6 are involved in the development of metabolic syndrome and insulin resistance. Although leptin is recognized for its ability to regulate appetite and metabolic rate, it also has proinflammatory and atherogenic effects that intensify the inflammatory environment in obesity [24].

Psoriasis patients, especially those exhibiting severe manifestations of the disease, are more likely to have metabolic syndrome, which includes obesity, insulin resistance, dyslipidemia, and hypertension [25]. Metabolic syndrome consists of a constellation of risk factors commonly linked to obesity, posing an elevated risk for atherosclerotic cardiovascular disease and type 2 diabetes. The National Heart, Lung, and Blood Institute (NHLBI) defines metabolic syndrome as follows: elevated fasting glucose (≥100 mg/dL or undergoing treatment for hyperglycemia), elevated blood pressure (≥130/85 mmHg or under antihypertensive treatment), reduced HDL cholesterol (<40 mg/dL in men, <50 mg/dL in women), hypertriglyceridemia (≥150 mg/dL or on treatment for hypertriglyceridemia), and abdominal obesity (waist circumference >102 cm in men or >88 cm in women). Currently, metabolic syndrome is outlined by the presence of three or more of these five criteria [26,27].

In the context of psoriasis, the correlation with metabolic syndrome is particularly interesting due to the interlinked inflammatory pathways that display noteworthy similarities, indicating a complex interplay that contribute to a damaging cycle [26]. This implies that the presence of one condition could exacerbate the pathophysiological mechanisms associated with the other [28].

Within patients affected by metabolic syndrome, there is a heightened production of proinflammatory mediators, such as leptin, TNF-α, and IL-6, creating a mutually reinforcing cycle that exacerbates both conditions [29]. Furthermore, the presence of metabolic syndrome in patients with psoriasis is linked to a more severe form of the disease, an increased likelihood of resistance to treatment, and a higher incidence of cardiovascular events [30].

The elucidation and comprehension of the pathophysiology of psoriatic disease and its underlying immune mechanisms has led to a groundbreaking transformation in the therapeutic approach to this condition. Recent advances in biological therapies and various inhibitors targeting the IL-23/IL-17 signaling axis have shown remarkable progress and efficacy, leading to significant improvements in patients with psoriasis [31]. Through this analysis of the immune pathways involved in psoriasis, therapeutic strategies have shown promising results in recent years, in particular with regard to biologics such as anti-IL-17, anti-IL-23, and anti-TNF-α. Nevertheless, methotrexate continues to maintain a prominent position among psoriasis treatments and remains generally used. Biologic therapies are designed to specifically target key cytokines in the Th17 inflammatory pathway. Anti-IL-17 efficiently blocks IL-17, which is a cytokine directly responsible for the recruitment of inflammatory cells and hyperproliferation of keratinocytes. Anti-IL-23 inhibits IL-23, which acts upstream and determines the differentiation and survival of Th17 cells. By reducing the downstream production of interleukin 17 and other inflammatory mediators, these therapies may provide early relief of clinical symptoms [32]. Anti-TNF-α has as its mechanism of action the neutralization of tumor necrosis factor alpha, a cytokine that is involved in chronic inflammation by activating keratinocytes and endothelial cells, contributing to the recruitment of T cells and the formation of psoriatic plaques [33]. Methotrexate, although providing a more generalized immunosuppressive effect, is less targeted than biologic agents; it acts by decreasing cytokine production and inhibiting keratinocyte proliferation [34].

The growing interest in understanding the impact of metabolic syndrome on the progression of psoriasis and the efficacy of treatments for both conditions arises from the significant challenges in simultaneously managing them [35,36].

The most widely used tool for assessing the severity and extent of psoriasis is the Psoriasis Area and Severity Index (PASI). This method involves evaluating four specific anatomical regions: the head, trunk, arms, and legs, which represent different proportions of the body surface area (BSA): the head (10%), upper limbs (20%), trunk (30%), and lower limbs (40%). The involvement of each region is graded on a scale from 0 to 6, where 0 indicates no involvement and 6 indicates full involvement. Moreover, PASI includes the evaluation of three clinical parameters: erythema, induration, and desquamation. The intensity of each parameter is scored on a scale from 0 to 4, with 0 indicating the absence of the condition and 4 indicating severe intensity. The final PASI score ranges from 0 to 72, with higher scores indicating more severe psoriasis [24].

Typically, PASI scores are determined before, during, and after treatment to assess the efficacy of therapy for psoriasis patients. Clinicians evaluate changes in absolute PASI and the percentage of PASI improvement, such as PASI 50, which signifies a 50% reduction in the PASI score from baseline. The introduction of highly effective therapies, such as biologics, has enabled many patients to achieve PASI 75 or even PASI 90 after treatment [37,38,39].

This study aims to evaluate the impact of metabolic syndrome on the efficacy of psoriasis treatments over a one-year period, using the PASI score as a measure of treatment response. The way comorbidities influence the management of psoriasis remains incompletely understood; thus, by analyzing the interaction between systemic inflammation, metabolic dysfunction, and treatment outcomes, we intend to improve the comprehension of the interaction between metabolic syndrome and psoriasis.

## 2. Materials and Methods

### 2.1. Patient Selection

A total of 150 individuals with clinically diagnosed psoriasis were included in this retrospective cross-sectional study conducted at the Department of Dermatology of the Emergency Clinical Hospital of Craiova between December 2022 and December 2023. The patients were classified into two subgroups: patients with psoriasis alone and patients with both psoriasis and metabolic syndrome. The study included 150 patients, with 38.67% women and 61.33% men. Participants were divided into two subgroups: 76 patients with psoriasis alone (PSO group) and 74 with both psoriasis and metabolic syndrome (PSO-MS group). The PSO-MS subgroup had a predominance of male patients (73%). Treatments were distributed as follows: anti-TNF-α (*n* = 52), anti-IL-17 (*n* = 32), anti-IL-23 (*n* = 46), and methotrexate (*n* = 20). Patients’ ages ranged from 26 to 76 years, categorized into three age groups: ≤49 years (31.33%), 50–59 years (33.33%), and ≥60 years (35.33%).

The inclusion criteria for patients were as follows: diagnosis of moderate to severe psoriasis, confirmed according to the American Academy of Dermatology guidelines, age exceeding 18 years, and the presence of metabolic syndrome based on the presence of at least three criteria set by the NHLBI and AHA: elevated fasting glucose > 100 mg/dL, or undergoing treatment for hyperglycemia, blood pressure > 130/85 mmHg, or undergoing treatment for arterial hypertension, low HDL-cholesterol levels (<40 mg/dL in men or <50 mg/dL in women) or under treatment for dyslipidemia, triglycerides > 150 mg/dL, or on treatment for hypertriglyceridemia and abdominal obesity (waist circumference > 102 cm in men or >88 cm in women). Exclusion criteria from the study were the presence of other autoimmune diseases unrelated to psoriasis, severe systemic diseases, such as advanced cardiovascular disease or end-stage chronic kidney disease, active or recent malignancies, and severe psychiatric conditions which could interfere with treatment adherence.

This research received approval from the Ethics Committee of the University of Medicine and Pharmacy of Craiova (approval number 195/20.09.2022), and all participants provided written informed consent. The study adhered to the principles outlined in the Declaration of Helsinki (2004).

### 2.2. Data Collection

For each patient, relevant demographic and clinical data were collected, including: gender, age, weight and abdominal circumference, presence of metabolic syndrome, total cholesterol, triglycerides, ASAT, ALAT, glycaemia, HDL-cholesterol, arterial hypertension (AHT), leptin, IL-17, IL-23, reactive C protein, neutrophiles, lymphocytes, and the PASI score.

Psoriasis was evaluated using the PASI score. For the ease of subsequent analysis, psoriasis was also classified into categories, based on the PASI score and the thresholds 5 and 10. Thus, three categories were created: 0 to 5: none to mild psoriasis, 6 to 10: moderate psoriasis, 11 or above: severe psoriasis [40].

### 2.3. Statistical Analysis

Statistical analysis performed following this investigation was conducted using Statistical Package for Social Sciences (SPSS) software, version 26.0 (IBM Corp., Armonk, NY, USA). Initially, detailed descriptive statistics and charts performed in Microsoft Excel were analyzed for the entire study group, offering a robust framework for further evaluations. Variables were expressed in terms of medians, or absolute and relative frequencies (%). Normality was assessed using the Shapiro–Wilk test. Possible associations and correlations between variables were assessed using the following statistical tools: Kendall’s Tau-b correlation; the Wilcoxon signed-rank test and Friedman test for repeated measures; and the Mann–Whitney U and Kruskal–Wallis H tests to determine if there were differences between the groups. For this study, the following *p* values were accepted: *p* < 0.05 significant in a confidence interval (CI) of 95%.

## 3. Results

The study cohort comprised 150 individuals diagnosed, at the beginning of this research, with psoriasis in different stages of the disease (mild = 0, moderate = 4, severe = 146). Almost half of these patients were previously diagnosed with metabolic syndrome (74 patients, representing 49.33% of the entire study cohort) and constituted the PSO-MS subgroup. The remaining 76 patients (50.67%) were included in the PSO subgroup.

From the entire study cohort, 38.67% were women, and 61.33% were men. The PSO subgroup included an equal number of females and males (38), but there was a clear predominance of males in the PSO-MS subgroup (54 males, representing 73% of all patients from this subgroup). A chi-square test for association was conducted between gender and subgroup. All expected cell frequencies were greater than five. There was a statistically significant moderate association between gender and subgroup, χ^2^(1) = 8.344, φ = 0.236, *p* = 0.004 (Table 1).

The youngest patient from the study cohort was 26 years old, and the eldest one was 76. Thus, three age groups were defined: patients with ages less than or equal to 49 years old (47 patients, representing 31.33% of the entire study cohort), patients with ages less than or equal to 59 years old (50 patients, 33.33%), and patients older than 60 years old (53 patients, 35.33%). Age groups were similar in terms of subgroup PSO-MS and PSO inclusion, and no statistical differences were identified between these categories, χ^2^(1) = 0.415, φ = 0.053, *p* = 0.813 (Table 1). On the other hand, females were mostly included in the first age group (44.8% of all females), in contrast with males that were mostly included in the last age group (42.4% of all males). Thus, there was a statistically significant association between gender and age groups, χ^2^(2) = 9.000, *p* = 0.011.

Patients included in the study group received four types of therapies: anti-IL-17, anti-IL-23, anti-TNF-α, and methotrexate. The distribution of the therapy types is similar for both study groups, *p* = 0.110 (Table 1).

According to the research methodology defined for this study, PASI level was evaluated for all subjects at the beginning of the study, then after 3 months, after 6 months, and after 1 year. During the 12-month period of research, variations were observed in all PASI categories, at all intermediate stages of evaluation (Table 2).

During the 1-year period of research, the overall number of patients with mild PASI levels increased for both the PSO and PSO-MS groups, mostly in accordance with a decreasing trend of patients within the severe group (Figure 1).

In the PSO subgroup, a significantly higher percentage reduction in the PASI score was therefore observed compared to the PSO-MS subgroup, which highlights the negative impact of metabolic syndrome on therapeutic success, emphasizing the slowing of psoriasis therapies through the interaction between systemic inflammation and metabolic factors.

Considering the overall tendency of PASI levels, the evolution of PASI between the main phases of the study was defined as the difference between the BG and 3MF PASI values, between the 6MF and 3MF values, and then between the 12MF and 6 MF values. Thus, a negative difference represented an increase of PASI level during that specific phase of the study, and a positive difference represented a decrease of the PASI level. The percentual variation at follow-up was computed as the ratio between the difference in PASI values and the initial values (expressed in absolute values).

For each phase of the study, PASI score variation and percentual variations were analyzed and tested for association with the other variables acquired within this study.

### 3.1. PASI Follow-Up After 3 Months

For all patients included in the study cohort, PASI values were measured at the beginning of the PSO treatment, and a first follow-up was performed 3 months later.

Since the variations in PASI scores in the 3-month interval were not normally distributed (*p* < 0.0005), a Wilcoxon signed-rank test was conducted to determine the effect of the PSO treatment on PASI values. The difference between PASI scores was approximately symmetrically distributed, as assessed by a histogram with a superimposed normal curve. Data are medians unless otherwise stated. Of the 150 participants recruited to the study, the PSO treatment elicited a decrease in PASI scores in 148 participants at follow-up. The most notable reduction in PASI score was observed in the first three months of treatment, which emphasizes the early efficacy of biologic treatments, especially anti-IL-23. Rapid improvements during this period are strongly associated with better long-term management of moderate to severe psoriasis.

There was a statistically significant median decrease in PASI values (11.80) before the treatment (17.10) to follow-up (5.80); z = −10.54, *p* < 0.001. The two patients with a negative percentual variation (−44%) initially started the PSO treatment with anti-TNF-α. Given the increase in PASI values 3 months after treatment initiation, the PSO treatment was changed to anti-IL-17.

PASI variation and percentual variation were determined for each therapy type (Figure 2).

The most rapid initial reduction was due to anti-IL-23 therapies, probably due to their upstream action in the inflammatory pathway, acting early by disrupting IL-17 production.

A Kruskal–Wallis test was conducted to determine if there were differences in PASI variations between groups defined by the therapy type: anti-TNF-α (*n* = 52), anti-IL-17 (*n* = 32), anti-IL-23 (*n* = 46), and methotrexate (*n* = 20). Distributions of PASI variations were similar for all groups, as assessed by visual inspection of the boxplots. Median PASI variations increased from 9.65 (anti-IL-17) to 11.0 (anti-IL-23), to 13.70 (methotrexate), up to 14.80 (anti-TNF-α), and there were statistically significantly differences between the different therapy types; χ^2^(3) = 7.983, *p* = 0.046 (almost borderline value). Subsequently, pairwise comparisons were performed using Dunn’s procedure with a Bonferroni correction for multiple comparisons. Adjusted *p*-values are presented. This post hoc analysis revealed no statistically significant differences in PASI variation values between therapy type groups (adjusted *p* > 0.05). On the other hand, PASI percentual variation was not statistically significantly different between therapy types (*p* = 0.102).

Similar tests (Mann–Whitney U) were performed to determine potential differences in PASI percentual variation for the study cohort (*p* = 0.05), gender (*p* = 0.799), and AHT presence (0.295). The only statistically significant difference was identified between study groups: the PSO group exhibited a median percentual variation of 71.50%, significantly higher than the median percentual variation of 68.00% for the PSO-SM group (U = 3334.00, z = 1.963, *p* = 0.050).

PASI variation during the first three months of the study was also analyzed according to the age group using a Kruskal–Wallis test. Distributions of both PASI variation and percentual variation were similar for all groups, as assessed by visual inspection of the boxplots. Median PASI percentual variations increased from 0.600 (age ≤ 49 years old) to 0.715 (50–59 years old), up to 0.730 (≥60 years old), and there were statistically significant differences between the different age groups; χ^2^(2) = 11.516, *p* = 0.003. Statistical analysis revealed significant differences in PASI percentual variations among the three defined age groups during the first three months of treatment. Pairwise comparisons confirmed that the youngest age group had significantly smaller percentual variations compared to both the middle age group (*p* = 0.004) and the oldest age group (*p* = 0.027). These findings suggest that older patients demonstrated more pronounced improvements during the initial stages of treatment.

Pairwise comparisons were performed using Dunn’s procedure with a Bonferroni correction for multiple comparisons. This post hoc analysis revealed statistically significant differences in median PASI score variation between the first age group and the third age group (*p* = 0.027) and between the first and second age group (*p* = 0.004), but not between the second and third. On the other hand, PASI variation was not statistically significantly different between age groups (*p* = 0.943).

Kendall’s Tau-b correlation was run to determine the relationship between PASI variations and BMI amongst the 150 participants. There was a moderate negative association between PASI percentual variation and BMI, which was statistically significant, τ_b_ = −0.135, *p* = 0.015. Thus, the higher the BMI was, the smaller the percentual variation recorded.

### 3.2. PASI Follow-Up After 6 Months

For all patients included in the study cohort, PASI values were also measured at a second follow-up stage, 3 months after the first follow-up, thus 6 months after the beginning of the study.

Variations in PASI values in this 3-month interval (so, between months 3 and 6) were not normally distributed (*p* < 0.0005), and so a Wilcoxon signed-rank test was conducted to determine the effect of the PSO treatment on PASI values. The difference between PASI values was approximately symmetrically distributed, as assessed by a histogram with a superimposed normal curve. Data are medians unless otherwise stated. Of the 150 participants recruited to the study, the PSO treatment elicited a decrease in PASI value in 140 participants at the second follow-up. There was a statistically significant median decrease in PASI values (8.50) after the first follow-up (5.80) to the second follow-up (2.60); z = −6.268, *p* < 0.0005. There were ten patients with a negative percentual variation, but their treatment was maintained.

PASI variation and percentual variation were determined for each therapy type (Figure 3).

A Kruskal–Wallis test was conducted to determine if there were differences in PASI variation between groups defined by the therapy type. Distributions of PASI variations were similar for all groups, as assessed by visual inspection of the boxplots. Median PASI variations increased from 7.60 (anti-IL-23) to 7.90 (anti-IL-17), to 9.55 (anti-TNF-α), up to 9.95 (methotrexate), and they were not statistically significantly different between the different therapy types; χ^2^(3) = 5.845, *p* = 0.119.

On the other hand, PASI percentual variation was statistically significantly different between therapy types (*p* = 0.016), as the median PASI percentual variations increased from 72% (anti-IL-23) to 75% (methotrexate), to 80.5% (anti-TNF-α), up to 90.5% (anti-IL-17). Pairwise comparisons were performed using Dunn’s procedure with a Bonferroni correction for multiple comparisons. This post hoc analysis revealed statistically significant differences in median PASI percentual variations between anti-IL-23 and anti-IL-17 (*p* = 0.008), but not between any other group combination.

In terms of gender, study group, and AHT presence, Mann–Whitney tests were performed to identify group differences. PASI score variation was not statistically significantly different between these categories (*p* > 0.05). But PASI percentual variation was higher in the PSO group, with a median value of 85%, compared to only 72% for the PSO-MS group (U = 3562.00, z = 2.822, *p* = 0.005). Similar values were obtained for comparing AHT status: patients with AHT experienced a reduced percentual variation compared to patients without AHT (72% compared to 85%); U = 3506.00, z = 2.628, *p* = 0.009.

PASI score and percentual variation between months 3 and 6 of the study were also analyzed according to age group using a Kruskal–Wallis test, and according to BMI using Kendall’s Tau-b correlation coefficient, but no statistically significant differences were identified (*p* > 0.05).

### 3.3. PASI Follow-Up After 12 Months

The final follow-up was performed after 12 months from the beginning of the study. Variations in PASI values in the final 6-month interval (between months 6 and 12) were not normally distributed (*p* < 0.0005), and so a Wilcoxon signed-rank test was conducted to determine the effect of the PSO treatment on PASI variation. The difference between PASI values was approximately symmetrically distributed, as assessed by a histogram with a superimposed normal curve. Data are medians unless otherwise stated. Of the 150 participants recruited to the study, the PSO treatment elicited a decrease in PASI value in 110 participants at the third follow-up. There was a statistically significant median decrease in PASI values (0.60) after the second follow-up (2.60) to the third follow-up (1.50); z = −5.000, *p* < 0.0005. There were 40 patients with a negative percentual variation, but their treatment was maintained.

PASI variation and percentual variation were determined for each therapy type (Figure 4).

A Kruskal–Wallis test was conducted to determine if there were differences in PASI variation between groups defined by the therapy type. Distributions of PASI variation were similar for all groups, as assessed by visual inspection of the boxplots. Median PASI score variation and percentual variation were negative for the group of patients who received methotrexate (−0.60 median score variation and −28% median percentual variation). For the other therapy types, PASI score variation increased from 0 (anti-IL-17) to 0.60 (anti-TNF-α), up to 1.80 (anti-IL-23), and there were statistically significantly differences between the different therapy types; χ^2^(3) = 9.510, *p* = 0.023. Pairwise comparisons were performed using Dunn’s procedure with a Bonferroni correction for multiple comparisons. Adjusted *p*-values are presented. The post hoc analysis revealed statistically significant differences in PASI score variation between the therapy type groups anti-IL-17 and methotrexate (adjusted *p* = 0.017). Similar results were obtained for the analysis of PASI percentual variation. An increasing trend was observed between the same groups, in the same order, and there were statistically significantly differences between therapy types (χ^2^(3) = 38.554, *p* < 0.0005). Pairwise comparisons performed using Dunn’s procedure with a Bonferroni correction for multiple comparisons revealed statistically significant differences between anti-IL-23 and all the other three groups (*p* < 0.05), but not between any other groups.

PASI score variation was similar for females and males, patients from PSO and PSO-MS groups, as well as for patients with and without AHT (*p* > 0.05). As observed in the previous phases of the study, patients within the PSO-MS subgroup experienced reduced percentual variation (median value 7%) compared to patients from the PSO subgroup (median value 57%) (U = 3838.00, z = 3.874, *p* < 0.0005). AHT presence also influenced the PASI percentual variation, with a median variation of 11% compared to 52.5% for patients without AHT (U = 3650.00, z = 3.181, *p* = 0.001).

During the final 6-month phase of the study, PASI score variation and percentual variation were similar for the three age groups (*p* > 0.05). Similar results were obtained for BMI analysis (*p* > 0.05).

### 3.4. PASI Evolution During the Entire Year of Study

Besides PASI indices, the following values were also analyzed at the end of the study time frame: cholesterol, triglycerides, ASAT, ALAT, HDL, and leptin.

As indicated in Table 3, not all parameters maintained statistically significant differences between groups following 1 year of PSO treatment. Cholesterol values decreased after 1 year, more for subgroup PSO-MS, resulting in more similar values and no more statistically significant differences between groups. TG, ASAT, and ALAT values also decreased for both groups, but their statistically significant differences were maintained. Leptin and HDL also maintained their original trend.

The PASI, the most significant indicator for PSO, showed an important decrease for both subgroups, reaching similar values closer to zero; thus, the statistically significant differences were no longer present. All patients with PSO responded very well to the treatment, regardless of whether they also had MS or not.

After 1 year of treatment, the same Kendall’s Tau-b correlation was run to determine the relationship between PASI level and the clinical parameters included in Table 4 amongst 150 participants. There were mostly weak to moderate, positive associations between the PASI index and the parameters included in Table 1, and most tests were statistically significant (Table 4). The initial associations were maintained only for ASAT, Leptin, and HDL.

The complete analysis of the entire study period (between the beginning and month 12) was based on a Friedman test used to determine if there were differences in PASI indices during the 12-month period. Pairwise comparisons (between the four different periods of time) were performed in SPSS with a Bonferroni correction for multiple comparisons. PASI scores were statistically significantly different at the different time points; χ^2^(3) = 321.615, *p* < 0.0005. The overall median PASI scores decreased from 17.1 at the beginning of the study to 5.8 at the 3-month follow-up, to 2.6 at the 6-month follow-up, and to 1.5 after 1 year. Post hoc analysis revealed statistically significant differences in PASI scores between all time periods (adjusted *p* < 0.05). While all therapies demonstrated sustained improvements, anti-IL-17 therapies achieved the most consistent long-term efficacy, reflecting their capacity to maintain low PASI scores over extended periods. This indicates that anti-IL-17 may provide superior disease control in chronic management scenarios compared to other treatments.

## 4. Discussion

According to several studies, metabolic syndrome and psoriasis frequently coexist [41,42,43,44]. The systemic inflammation linked to psoriasis, characterized by elevated cytokines such as TNF-α, IL-6, and IL-17, raises the likelihood of metabolic imbalances [45].

The introduction of biologic therapies has revolutionized autoimmune disease treatments, especially for patients with moderate to severe forms of psoriasis. Recent research has shown that biologic therapies have the potential to reduce inflammatory markers, having a significant role in the management of comorbidities associated with chronic diseases [44]. The presence of metabolic syndrome, along with its components such as obesity, has the potential to reduce the efficacy of biologic treatments for psoriasis [45,46,47]. For instance, a multicenter study involving 262 patients that were previously diagnosed with psoriasis found that patients without metabolic syndrome were more likely to show positive responses to biologic therapies compared to those with associated metabolic syndrome [48].

The choice of treatment for patients suffering from metabolic syndrome in addition to psoriasis can become challenging and is influenced by several factors. For example, in the patient suffering from arterial hypertension, AHT can alter both the pharmacokinetics and pharmacodynamics of biologic therapies, influencing the processes of absorption, distribution, metabolism, and excretion of therapies, leading to suboptimal or inconsistent drug concentrations [49]. Similarly, insulin resistance can greatly diminish the effectiveness of psoriasis treatments by maintaining a continuous inflammatory cycle [50].

In the current study, we aimed to evaluate the impact of metabolic syndrome on various psoriasis treatments by using the PASI score across several time intervals: during the first 3 months, 3 to 6 months, and 6 to 12 months. In general, the results revealed that all therapies were efficient in psoriasis management, as suggested by the important PASI score reductions. Still, the improvement in symptomatology and the degree of reduction in PASI score varied significantly, both between the different types of treatments and between the PSO and PSO-MS subgroups.

All patients included in the study experienced a decrease in PASI score within one year. According to the results, the initial median PASI of 17.1 decreased to 5.8 at 3 months, 2.6 at 6 months, and finally to 1.5 at 12 months, demonstrating that the treatments administered were effective in both patient subgroups. However, within the PSO subgroup, there was a significantly higher percentage improvement in PASI reduction compared to the PSO-MS subgroup, implying that metabolic syndrome may reduce the efficacy of therapy, potentially due to the inflammatory and metabolic disturbances it causes, therefore complicating the pathophysiology of psoriasis.

In terms of the types of therapy studied, our research showed that patients who received anti-IL-17 had the best response in decreasing the PASI score during one year of treatment, with visible results at the end of the first 6 months, when the greatest improvement in median PASI was achieved, reaching an impressive 90.5%.

During the first 3 months of treatment, there was the most significant reduction in PASI scores, which is of significant importance, as a drastic reduction in PASI scores in the first few months usually indicates effective long-term disease management. Feldman et al. [51] found that achieving a reduction of PASI 40 or more during the first weeks of treatment was a strong predictor of long-term success in the management of moderate-severe psoriasis. During this phase, we noted that, regarding therapy type, anti-IL-23 was associated with the largest PASI reduction, with 52.2% of patients reaching a mild PASI score. In the development of psoriasis, IL-23 plays a crucial role by supporting the survival, proliferation, and activity of Th17 cells, which are essential in sustaining long-lasting inflammatory responses. When IL-23 binds to its receptor on these cells, JAK2 and STAT3 signaling pathways are being activated. This activation leads to the secretion of several proinflammatory cytokines, especially IL-17A and IL-17F [52,53,54]. These cytokines are crucial in the development of psoriasis. They both stimulate the proliferation of keratinocytes and promote the secretion of several chemokines, as well as antimicrobial peptides like b-defensin [55,56]. This cascade of events leads to the infiltration of neutrophils and other inflammatory cells into psoriatic lesions, thereby amplifying the inflammatory response. IL-23 plays a significant role in Th17 cells and also affects other immune cells, including macrophages and dendritic cells [3,56,57].

The prompt response noted with anti-IL-23 aligns with recent studies indicating that IL-23 inhibitors can effectively disrupt the inflammatory cycle in specific cases, providing rapid symptom relief for patients suffering from severe conditions, including psoriasis [58,59,60]. A meta-analysis of 13 studies examining the pooled data from all IL-23 inhibitors demonstrated an important class effect in efficacy when compared to both placebo and adalimumab. Specifically, the pooled relative risk (RR) of achieving a 75% improvement in PASI score was found to be 11.5 for IL-23 inhibitors compared to a placebo, and 1.9 for achieving a complete clearance (PASI 100) with IL-23 inhibitors versus adalimumab. The 95% confidence intervals (CIs) for these results were 9.4–13.9 for PASI 75 and 1.5–2.2 for PASI 100 [61].

However, in a study conducted by Yao and Lebwohl in 2019 [62], IL-17 inhibitors were the fastest to reach PASI 75, which is in contrast with our findings. In addition, Egeberg et al. [63] revealed that anti-IL-17 therapies acted faster and were more effective in achieving substantial results compared to IL-23 inhibitors. One possible explanation for this discrepancy could be the difference in patient characteristics or baseline disease severity between study populations.

Over the 3–6 month treatment period, anti-IL-23 therapy maintained its efficacy in reducing PASI in patients with moderate to severe psoriasis. As numerous studies suggest, IL-23 inhibitors have the best safety profile and may provide sustained control of psoriasis symptoms due to their high rates of response [64]. The significant decreases in median PASI score in our research indicated that the therapeutic effects that were evident in the initial phase continued in the intermediate phase, exhibiting a consistent therapeutic response, with most patients presenting either continued improvement or stable disease control without major fluctuations in PASI score. Although improvements were still evident, the rate of change in PASI score reduction started to enter a plateau phase, indicating stabilization of the therapeutic response. This suggests that even if most of the inflammatory factors have been addressed, immune activation may persist, and therefore ongoing treatment is necessary for maintenance. In the 6–12 month period, a median PASI variation of 1.80 was observed, suggesting a steady state of improvement and stabilization, controlling psoriatic symptoms, with slightly further improvement in some patients. This relatively low variation in the PASI indicates that there were no significant fluctuations, with the majority of patients showing either modest improvement or stable PASI scores over this time interval.

Anti-IL-17 showed a remarkable efficacy, especially in the long term, although the initial response was slower than anti-IL-23. In the first 3 months, anti-IL-17 had a sustained positive response on the PASI, but with a lower rate of reduction than anti-IL-23 and anti-TNF-α. Direct inhibition of IL-17, a key cytokine in the inflammatory process, led to durable results and stable maintenance of the PASI at reduced levels. IL-17 stimulates keratinocyte proliferation and the attraction of neutrophils and other inflammatory cells to psoriatic lesions, and so blocking this cytokine has a direct and sustained effect on keratinocyte proliferation, but requires a longer period of time to produce significant decreases in PASI score [65]. The greatest percentage change in PASI score was achieved over the 3–6 month period under anti-IL-17 treatment, demonstrating that anti-IL-17 has a superior ability to provide stable control of inflammation and psoriatic symptoms. In the long term, anti-IL-17 proved to be more effective in maintaining a low PASI score, with patients managing to keep symptoms under control. Due to its direct action on skin inflammation, anti-IL-17 is ideal for patients with chronic psoriasis requiring long-term treatment to keep the disease under control [66]. This difference in outcomes can be attributed to the specific mechanisms of action employed by each treatment. IL-23 functions upstream in the Th17 signaling pathway, whereas the direct inhibition of IL-17 leads to more significant and lasting reductions in skin inflammation. This is due to IL-17 acting as a direct effector cytokine within the inflammatory process [67].

Anti-TNF-α was one of the first biologic therapies used in psoriasis. TNF-α inhibitors interrupt the inflammatory cycle by neutralizing TNF-α, providing both rapid and sustained relief of symptoms associated with severe psoriasis. TNF-α is a cytokine produced by the immune cells and regulates inflammatory responses by binding to TNF-receptor 1 (TNFR1), which is proinflammatory, and TNF-receptor 2 (TNFR2), which is anti-inflammatory. The balance between these receptors regulates cell functions and recruits inflammatory cells. An upregulation of TNFR1 and TNFR2 was observed in the dermal blood vessels of psoriatic lesions [68,69,70].

In our study, anti-TNF-α has also been shown to be effective in rapidly reducing the PASI, making it a viable solution for patients who require an immediate reduction in inflammation and improvement of symptoms in a short time. Its mechanism of action, by inhibiting TNF-α, prevents the activation of T cells and the release of proinflammatory cytokines and chemokines, having a rapid effect on chronic inflammation in psoriasis [71,72]. Although efficient in the first months of treatment, as the study progressed, anti-TNF-α efficacy began to decline, and by 6 months and especially at 12 months, this therapy was less durable compared with anti-IL-17 and anti-IL-23.

Methotrexate also provided visible results in reducing the PASI score; these results were modest, especially compared to anti-IL-17 and anti-IL-23. By the end of one year, biologic therapies had clearly surpassed methotrexate, particularly in terms of achieving higher levels of PASI improvement, such as PASI 90.

Regarding the impact of metabolic syndrome on treatment response, we identified a statistically significant association (*p* = 0.015). Metabolic syndrome is a constellation of medical conditions, including hypertension, obesity, dyslipidemia, and insulin resistance, all of which share systemic inflammation as a common feature. Adipose tissue, especially visceral adipose tissue, is highly immunologic and endocrine-active. It produces and releases proinflammatory cytokines that exacerbate the inflammatory response, and this over-activation of the immune system may further contribute to reduced efficacy of psoriasis treatments [73]. In our study, we highlighted that as BMI increases, the percentual variation in PASI tends to be smaller, meaning that people with a higher BMI had, on average, a lesser decrease in the severity of psoriasis symptoms compared to those with a lower BMI. Obese individuals may experience a change in the pharmacokinetics and pharmacodynamics of psoriasis treatments, which may lead to lower drug exposure, and, therefore, to a decreased therapeutic response [74]. Immune system activity in psoriasis is mediated by T cells, particularly the Th1 and Th17 subpopulations. Studies indicate that obesity may amplify the activity of Th17 cells, thereby increasing the severity of psoriasis symptoms and decreasing the therapeutic effects of various therapies administered to these patients [74,75].

The percentage of patients who achieved a PASI reduction of more than 75% was up to 30% lower among patients with a higher BMI than among patients with a normal BMI, according to a cohort study conducted in Italy [76]. Moreover, Di Lernia et al. [77] proposed increased BMI as a separate predictor of treatment failure in a retrospective observational study including 110 patients. According to the study, only 42.21% of patients continued to receive the same treatment after two years.

Obesity is associated with a 60% greater likelihood of poor response to treatment in patients with psoriasis compared to patients with a normal BMI, as has been repeatedly demonstrated in several scientific research studies. A meta-analysis including 54 studies and 19,372 patients supported these findings. For each unit increase in BMI, there is a 6.5% increase in the odds of inadequate response to treatment [78]. A BMI greater than 30 is linked to increased psoriasis activity at baseline, as well as decreased response to therapy and treatment adherence (HR:1.85), according to an observational cohort study conducted in Denmark [79]. Another reason why obese individuals may respond more poorly to treatment could be oxidative stress, which plays a significant role in the severity of psoriasis, particularly in the context of obesity. Oxidative stress contributes to chronic inflammation by increasing ROS production in the keratinocytes of psoriatic lesions, activating inflammatory pathways such as NF-kB and MAPK, favoring cell proliferation and psoriatic plaque formation [79,80,81,82]. In obese people, excess adipose tissue intensifies systemic oxidative stress, which aggravates skin inflammation and can make psoriatic lesions more resistant to treatment [83].

Furthermore, as we analyzed the factors influencing therapeutic response in psoriasis, we observed that age seemed to play an essential role in the rate of reduction of the PASI score, especially in the initial stages of treatment. Patients aged 50–59 years showed more rapid decreases in PASI in the first 3 months of treatment compared to the other two subgroups; this difference could have a rationale related to immune system function and inflammatory response, as aging may influence cytokine levels and immune cell activity, which could facilitate a more rapid suppression of inflammatory processes once treatment is started. Hormonal fluctuations, particularly in patients in the adolescent–young adult range, may also serve as an explanation. Cortisol, estrogen, and testosterone can influence inflammatory pathways through immunomodulatory effects and may interfere with the response to treatment by hyperreactivity of the immune system [84]. In our research, at the end of the 12 months, almost 80% of patients under 49 were still in the category of mild psoriasis, while 21.3% remained with moderate forms of the disease. These results contrast with prior studies, which indicate that younger aged patients tend to have a faster and more effective response, most likely due to a more robust immune system and a superior general health status without significant comorbidities [85]. This discrepancy could be explained by the particularities of chronic inflammation in psoriasis associated with an early onset of the disease, influenced by a genetic component predisposed to a marked and persistent inflammatory response. Therefore, these findings indicate that the variability of response to treatment depends not only on age but also on the genetic and pathophysiologic particularities of each individual’s disease.

Gender did not demonstrate a statistically significant direct impact on PASI score reduction overall, but sex-related trends were evident in the characteristics and severity of psoriasis at baseline, particularly in MS patients. Men with MS generally had more severe forms of the disease, which could be attributed to lifestyle factors including higher rates of smoking, alcohol consumption, and obesity. These factors may exacerbate systemic inflammation, contributing to more pronounced psoriasis symptoms in men [86,87].

Leptin, as we have shown in our study, appears to be involved not only in metabolic health [88], but also in the pathophysiology of psoriasis [89], especially in those cases complicated by MS. Over the one-year period, we observed a strong correlation between leptin and PASI scores, suggesting a role in the maintenance or even worsening of the chronic inflammation characteristic of psoriasis. Given the observed association between leptin levels and PASI scores, several markers of lipid metabolism such as leptin could serve as potential biomarkers for the management of patients with PSO-MS. This association between leptin and the PASI is significantly influenced by the proinflammatory characteristics of leptin. The adipose tissue secretes leptin, which stimulates the production of proinflammatory cytokines and induces keratinocyte proliferation, ultimately leading to the formation of psoriatic plaques [88,89,90,91,92]. Elevated leptin levels may exacerbate pre-existing inflammation and hinder the therapeutic response, which may explain why patients with higher leptin levels had a less pronounced decrease in PASI scores throughout our study. Future studies should assess whether incorporating leptin and other markers of lipid metabolism into clinical protocols could help tailor treatments more effectively, particularly for patients with severe comorbidities.

Consideration should also be given to how environmental influences, such as metabolic syndrome and arterial hypertension, can affect psoriasis patients and amplify the symptoms of the disease. These factors contribute to exaggerated inflammation and an excessive immune response. Moreover, recent research suggests that the COVID-19 pandemic also contributed negatively to the vulnerability of psoriasis patients, as SARS-CoV-2 infections and COVID-19 vaccines caused inflammatory reactions that exacerbated psoriasis symptoms. However, vaccination is still recommended for patients with psoriasis, as the protective benefits against COVID-19 outweigh the risks. It is important that patients receive a complete disease management approach, including alleviating systemic inflammation and treating environmental factors that contribute to disease progression [93].

One of the primary limitations of the study was the small sample size of patients, who were divided into several subgroups based on type of therapy, the presence of metabolic syndrome, and other factors, the effective sample size for each subgroup becoming relatively small, which may have influenced the statistical power of the study, potentially affecting the reliability of comparative analyses between treatments. In addition, the study population was focused on specific demographic characteristics, such as the majority of male patients among the PSO-MS subgroup, which may not fully represent the broader psoriasis patient population. As such, the results may not be universally applicable to different demographic groups or populations with different ethnic, age, or gender compositions. A further limitation of the study might be the one-year follow-up period; although it provides particularly important information on short- and medium-term treatment efficacy, data on the long-term durability of these effects are lacking.

## 5. Conclusions

In summary, within the limitations of this study, we have demonstrated the need to understand the interaction between psoriasis and metabolic syndrome in modeling the outcomes of different types of treatments administered. Metabolic syndrome significantly impacts psoriasis and specific therapies. Although all treatments showed improvements in PASI score over one year, patients who had both psoriasis and metabolic syndrome showed comparatively diminished improvements, suggesting the need for a therapeutic approach that targets both dermatologic and systemic factors. Among the therapies examined, biologics, especially anti-IL-17, showed the most significant long-term efficacy. In addition, elevated BMI and serum leptin levels were acknowledged as critical factors associated with delays in therapeutic responses, further positioning the negative role of systemic inflammation and metabolic imbalances in the management of psoriasis.

Although the results obtained provide valuable insights, further research, such as multicenter, long-term studies, is needed to validate these findings, but also to improve treatment strategies in the management of these conditions that are intertwined by pathophysiologic mechanisms.

## Figures and Tables

**Figure 1 diagnostics-14-02887-f001:**
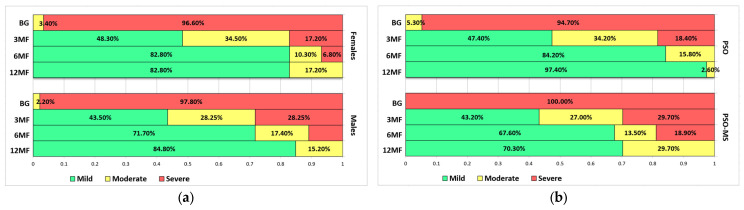
Evolution of PASI scores at BG, 3MF, 6MF, and 12MF, distributed by (**a**) gender; (**b**) study group; (**c**) age group; and (**d**) therapy type.

**Figure 2 diagnostics-14-02887-f002:**
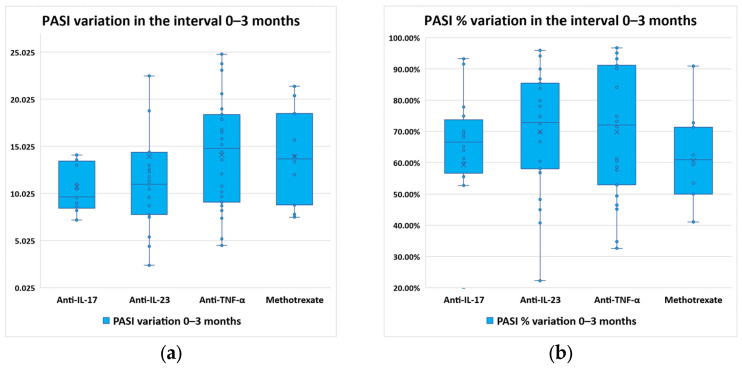
Evolution of PASI scores in the first three months of the study, distributed by therapy type: (**a**) score variation; (**b**) percentual variation.

**Figure 3 diagnostics-14-02887-f003:**
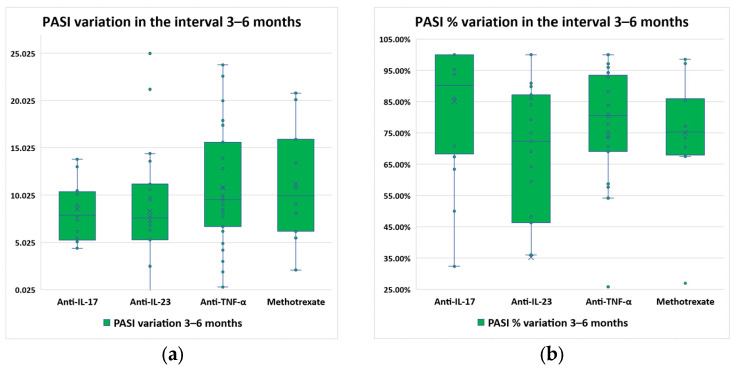
Evolution of PASI scores between month 3 and month 6 of the study, distributed by therapy type: (**a**) score variation; (**b**) percentual variation.

**Figure 4 diagnostics-14-02887-f004:**
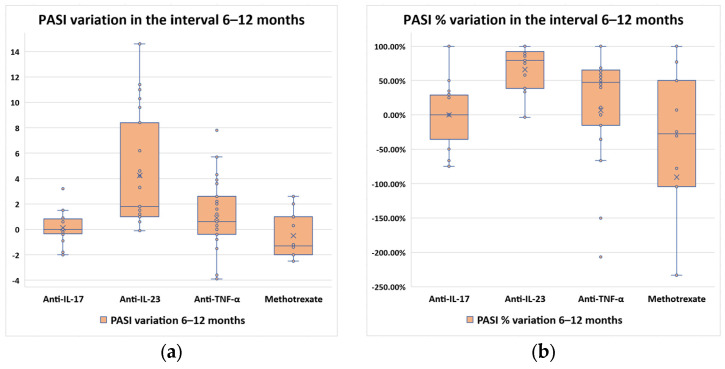
Evolution of PASI scores between month 6 and month 12 of the study, distributed by therapy type: (**a**) score variation; (**b**) percentual variation.

**Table 1 diagnostics-14-02887-t001:** Demographic characteristics of the study groups.

Parameter	Values	PSO-MS	PSO	Total	*p*
		74 Patients	76 Patients	150 Patients	
Gender	F	20 (34.48%)	38 (65.52%)	58 (38.67%)	0.004 *
		27.03%	50%		
	M	54 (58.7%)	38 (41.3%)	92 (61.33%)	
		72.97%	50%		
Age group (years old)	≤49	22 (46.81%)	25 (53.19%)	47 (100%)	0.813 *
		29.73%	32.89%		
	50–59	24 (48%)	26 (52%)	50 (100%)	
		32.43%	34.21%		
	≥60	28 (52.83%)	25 (47.17%)	53 (100%)	
		37.84%	32.89%		
Therapy type	Anti-IL-17	16 (50%)	16 (50%)	32 (100%)	0.110 *
		21.62%	21.05%		
	Anti-IL-23	24 (52.17%)	22 (47.83%)	46 (100%)	
		32.43%	28.95%		
	Anti-TNF-α	20 (38.46%)	32 (61.54%)	52 (100%)	
		27.03%	42.11%		
	Methotrexate	14 (70%)	6 (30%)	20 (100%)	
		18.92%	7.89%		

* Chi-square test. Values emphasized in light grey represent the sum of values by column.

**Table 2 diagnostics-14-02887-t002:** PASI index evolution during the main phases of the study.

Stage	PASI Index—*n* (%)					
	PSO-MS Group			PSO Group		
	Mild	Moderate	Severe	Mild	Moderate	Severe
Beginning (BG)	0 (0.0%)	0 (0.0%)	74 (100.0%)	0 (0.0%)	4 (5.26%)	72 (94.74%)
Follow-up at 3 months (3MF)	32 (43.24%)	20 (27.03%)	22 (29.73%)	36 (47.37%)	26 (34.21%)	14 (18.42%)
Follow-up at 6 months (6MF)	50 (67.57%)	10 (13.51%)	14 (18.92%)	64 (84.21%)	12 (15.79%)	0 (0.0%)
Follow-up at 12 months (12MF)	52 (70.27%)	22 (29.73%)	0 (0.0%)	74 (97.37%)	2 (2.63%)	0 (0.0%)

**Table 3 diagnostics-14-02887-t003:** Main characteristics of the study group after 1 year.

Parameter	After the Diagnosis			After 1 Year		
	PSO-MS	PSO	*p*	PSO-MS	PSO	*p*
	Median	Median		Median	Median	
Cholesterol	230.63	219.00	0.026	212.100	205.090	0.299
TG	238.00	123.55	<0.0005	186.000	95.645	<0.0005
ASAT	21.00	19.20	0.845	19.600	21.300	0.916
ALAT	26.90	21.10	0.007	24.100	20.250	0.014
Leptin	1131.089	481.183	<0.0005	744.621	299.311	<0.0005
PASI	21.00	14.90	0.029	2.000	1.200	0.209
HDL	36.00	55.62	<0.0005	50.000	58.000	<0.0005

**Table 4 diagnostics-14-02887-t004:** Kendall’s Tau-b correlation between the PASI and all clinical parameters, following 1 year of treatment.

	After the Diagnosis		After 1 Year	
	Tau-b Coefficient τ_b_	*p*	Tau-b Coefficient τ_b_	*p*
Cholesterol	0.226	<0.0005	0.061	0.288
TG	0.221	<0.0005	0.104	0.069
ASAT	0.120	0.032	0.126	0.028
ALAT	0.043	0.439	0.180	0.002
Leptin	0.191	0.001	0.322	<0.0005
HDL	−0.134	0.016	−0.117	0.043

## Data Availability

The authors declare that the data of this research are available from the corresponding authors upon reasonable request.

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
