# Peer review of "The Role of Metabolic Syndrome in Psoriasis Treatment Response: A One-Year Comparative Analysis of PASI Progression"

_diagnostics, 2024, doi:10.3390/diagnostics14242887_

Round 1
Reviewer 1 Report
Comments and Suggestions for Authors
Dear Authors,
I have already carefully review the manuscript titled “The Role of Metabolic Syndrome in Psoriasis Treatment Response: A One-Year Comparative Analysis of PASI Progression”. This manuscript explores the role of metabolic syndrome in the treatment response of psoriasis over a one-year period, using PASI scores to assess the efficacy of various treatment modalities. The study provides valuable insights into the complex interplay between metabolic syndrome, systemic inflammation, and psoriasis treatment outcomes. While the manuscript is well-structured and presents detailed findings, I believe certain revisions can enhance its clarity and impact. Below, I outline my major recommendations.
1. Include Psoriasis Treatment Overview in Introduction:
Although the focus of this study is on metabolic syndrome's influence on psoriasis treatment, the Results and Discussion sections discuss four treatment modalities (anti-IL-17, anti-IL-23, anti-TNF-α, and methotrexate) in detail. Adding a brief overview of these treatment options in the Introduction would provide essential context for readers, particularly those less familiar with psoriasis therapies. This addition would also establish a better foundation for the subsequent comparison of treatment outcomes.
2. The Introduction currently focuses on the background of psoriasis and its association with metabolic syndrome but lacks a clear transition to the specific aims of this study. I suggest concluding the Introduction with a paragraph summarizing the study's objectives and its broader significance. This will help the Introduction flow more cohesively and provide readers with a clear understanding of what the study seeks to accomplish.
3. Materials and Methods: The "Patient Selection" section could benefit from greater detail. For instance, including the demographic distribution (e.g., gender ratios) and the number of patients receiving each treatment would improve the clarity and transparency of the methodology. While this information is available in the Results section, presenting it here would make the manuscript easier to follow and align with standard reporting practices.
4. The Results section is comprehensive and provides detailed descriptions of the findings. However, it mainly focuses on observations, such as correlations and statistical differences, without fully connecting them to the study’s broader conclusions. For example, while it is clear that there are significant differences between groups, it is less apparent why these differences matter in the context of the study's aims.
To address this, I recommend incorporating some of the interpretative and conclusion-oriented statements currently found in the Discussion section into the Results. For instance, lines 402–404, 410–413, and 414 in the Discussion provide excellent analyses, such as the observation that “During the first three months of treatment, there was the most significant reduction in PASI scores.” Including such statements in the Results section would improve its logical flow and help readers better understand the significance of the findings. Importantly, this does not mean removing these analyses from the Discussion but rather enhancing the Results to provide a more cohesive narrative.
Best,
Author Response
Thank you very much for revising our manuscript. Here are the detailed responses below:
- Include Psoriasis Treatment Overview in Introduction: Agree. We have, accordingly, added a brief summary of the mechanisms of action for biologics in the Introduction section. (please find it at line 120-136).
- The Introduction currently focuses on the background of psoriasis and its association with metabolic syndrome but lacks a clear transition to the specific aims of this study: Thank you for pointing this out. We have revised the final paragraph of the introduction to outline the study's aims. (line 156-161)
- Materials and Methods: We have modified as suggested (line 168-175)
- The Results section: we have added some interpretative and conclusion-oriented statements: line 253-257, line 276-279, line 289-291, line 451-455.
Reviewer 2 Report
Comments and Suggestions for Authors
i read with great interest the manuscript
comments
abstract:
- line 28 report briefly the vicious circle of MS and obesity and psoriasis
- explain the abbreviation PSO ( meaning psoriasis) or better non-MS psoriasis patients line 33 or introduce the categories before this report
intro
-report more on the pathophysiology link - vicious circle between obesity and MS and psoriasis( along with cardiometabolic cormobidities)
-report more about the criteria and categories of MS
methods- well presented
results
- maybe the therapy type should be in a sepatated table as it is the focus of the manuscript
-was any statistical correlations in age and gender category
-general nice presentation of results
discussion
- extensive and nice discussion on the topic
some more points that are needed to be briefly reported in discussion or other points of the manuscript
-- should lipid metabolism markers be used as biomarkers in case of treating a MS-PSO patient
- Treatment choice is influenced by other parameters of MS- such as hypertension and insulin resistance ( so more confounders need to be assessed in each case)
- the whole exposome of the patients including comorbidities such as metabolic syndrome and hypertension or ageing and gender etc makes the patient more vulnerable to psoriasis flare when a trigger occurs due to the inflammation contribution (10.3390/vaccines12020178)... this disturance of balance should also be reported
Author Response
Thank you very much for taking the time to review this manuscript and for your thoughtful and constructive feedback. We have carefully considered each of your points and have incorporated the suggested changes to enhance the manuscript. Below is a summary of how we have addressed your comments:
Abstract:
- Line 28 - Vicious Circle of MS, Obesity, and Psoriasis: We have reported the vicious cycle of metabolic syndrome, obesity, and psoriasis in the abstract, underscoring their interconnected inflammatory pathways and their impact on treatment outcomes.
- Abbreviation PSO: The abbreviation "PSO" (psoriasis) has been clarified, and the distinction between non-MS psoriasis patients (PSO) and psoriasis patients with metabolic syndrome (PSO-MS) is introduced earlier for better comprehension.
Introduction:
-report more on the pathophysiology link - vicious circle between obesity and MS and psoriasis( along with cardiometabolic cormobidities) : We agree with this comment. We expanded the discussion on the pathophysiology of metabolic syndrome and its connection to psoriasis. This now includes detailed descriptions of shared inflammatory pathways between psoriasis and MS. (LINE 81-92)
-report more about the criteria and categories of MS: Thank you for pointing this out. We provided a detailed examination of the diagnostic criteria for metabolic syndrome as established by the NHLBI and the AHA. LINE 95-103
Methods: Thank you for the positive feedback on this section.
Results:
-maybe the therapy type should be in a sepatated table as it is the focus of the manuscript: - Table 1 was inserted in the manuscript. This table emphasized the distribution of gender, age groups and therapy type for the two study groups – PSO-MS and PSO.
-was any statistical correlations in age and gender category: -- A correlation between gender and age groups is analyzed in lines 233-238.
Discussion:
-- should lipid metabolism markers be used as biomarkers in case of treating a MS-PSO patient: Thank you for pointing this out. We have integrated the potential role of lipid metabolism markers, especially leptin, as biomarkers for guiding the need for personalized therapeutic strategies. (LINE 655-657, LINE 664-666).
- Treatment choice is influenced by other parameters of MS- such as hypertension and insulin resistance ( so more confounders need to be assessed in each case): Thank you for this suggestion. We have provided additional details. (lines 470-476)
- the whole exposome of the patients including comorbidities such as metabolic syndrome and hypertension or ageing and gender etc makes the patient more vulnerable to psoriasis flare when a trigger occurs due to the inflammation contribution (10.3390/vaccines12020178)... this disturance of balance should also be reported. RESPONSE: Agree. We have added additional information and the reference (lines 667-677)
Round 2
Reviewer 2 Report
Comments and Suggestions for Authors
The authors did take my suggestions into account and the manuscript was improved Well done